# Effect of Light-Emitting Diodes and Ultraviolet Irradiation on the Soluble Sugar, Organic Acid, and Carotenoid Content of Postharvest Sweet Oranges (*Citrus sinensis* (L.) Osbeck)

**DOI:** 10.3390/molecules24193440

**Published:** 2019-09-22

**Authors:** Linping Hu, Can Yang, Lina Zhang, Jing Feng, Wanpeng Xi

**Affiliations:** 1College of Horticulture and Landscape Architecture, Southwest University, Chongqing 400716, China; 18875065318@163.com (L.H.); yangcan0929@hotmail.com (C.Y.); zhangln0923@163.com (L.Z.); 15520104087@163.com (J.F.); 2Key Laboratory of Horticulture Science for Southern Mountainous Regions, Ministry of Education, Chongqing 400715, China

**Keywords:** light, UV, soluble sugar, organic acid, carotenoid, sweet orange

## Abstract

Mature ‘Hamlin’ sweet oranges (*Citrus sinensis* (L.) Osbeck) were irradiated using light-emitting diodes (LEDs) and ultraviolet (UV) light for six days after harvest. Based on evaluation of the basic ripening parameters of fruits, the contents of soluble sugars, organic acids, and carotenoids were analyzed (in pulps) on the sixth day by high-performance liquid chromatography (HPLC). The results showed that LED and UV irradiation not only accelerated orange ripening but also caused significant changes in the soluble sugar, organic acid, and carotenoid content. Compared with fruit subjected to dark shade (DS) treatment, the total soluble sugar, fructose, and glucose contents increased significantly in UV-treated (UVA, UVB, and UVC) fruits, while the sucrose content increased remarkably in white light, UVB, and UVC-treated fruits (*p* < 0.05). UV treatment was associated with inducing the largest effect on the total soluble sugar content. Except for UVB, other types of light notably induced an accumulation of the total organic acid content, none but blue light and red light markedly induced citric acid accumulation (*p* < 0.05). Interestingly, only the red light and dark shade treatments had markedly positive effects in terms of inducing carotenoid accumulation, including the total carotenoid, isolutein, zeaxanthin, lutein, neoxanthin, all-trans-violaxanthin, phytofluene, cis-ζ-carotene, and β-carotene concentrations. Other light treatments had significantly negative effects on carotenoid accumulation (*p* < 0.05). Therefore, soluble sugar, organic acid, and carotenoid accumulation in sweet oranges vary depending on the levels of UV and LED irradiation. Appropriate light irradiation is a potentially effective way to maintain or improve postharvest fruit quality.

## 1. Introduction

Citrus is a staple type of crop worldwide; their fruits not only have a unique flavor but also contain rich concentrations of health-promoting phytochemicals—such as phenolics, carotenoids, and vitamin C—that exhibit antioxidant, anti-inflammatory, antibacterial, and anticancer activities [1]. Sweet oranges (*Citrus sinensis* (L.) Osbeck) account for approximately 60% of citrus production for both fresh fruit and processed juice consumption, furthermore, its fruit and juice are important sources of human nutritional value [2]. Similar to other fruits, citrus quality will decline gradually after harvest, seriously reducing its edible and commodity value [3,4].

Many studies have investigated the role of postharvest exogenous chemical treatment on the maintenance of fruit quality. However, light irradiation is more attractive than chemical methods because it is residue-free and safe. Thus far, some advances have been achieved in determining the influence of light on fruit quality and related metabolites [5,6]. For example, ultraviolet C (UVC) and pulsed light treatments enhance the antioxidant properties of tomatoes [7], light-emitting diode (LED) irradiation promotes the accumulation of anthocyanins in sweet cherries [7,8], LED treatment induces ripening and improves the nutritional quality of postharvest banana fruit [9], and intense white light influences the carotenoid accumulation of developmental bilberry fruit, specifically responding to red/far-red light [10]. Even so, most of these studies only focus on investigating the effect of monochromatic light on fruit quality and related metabolites; few publications have reported on the use of polychromatic light, especially LED and UV light. Previous studies have shown that blue light treatment is beneficial for the accumulation of carotenoids in citrus fruits [11,12], suggesting the potential regulatory role of light irradiation on fruit quality-related metabolites. UV and LED irradiation have different effects on many aspects of fruit growth and development [13], but an understanding of the effects of postharvest UV and LED irradiation on the accumulation of fruit quality-related metabolites remains very limited.

Soluble sugars, organic acids, and carotenoids are important components that determine organoleptic qualities such as taste and color. Most importantly, carotenoids, which are also major health-promoting compounds, have a strong impact on the nutritional quality of sweet oranges and are representative indicators of fruit quality [3,14,15]. ‘Hamlin’, an important sweet orange cultivar in production [16], is widely loved by consumers for its moderate sweetness to acidity ratio and graceful color [17].

This work aims to evaluate the effect of postharvest LED (dark shade, DS; red light, RL; blue light, BL; and white light, WL) and ultraviolet A, B, and C (UVA, UVB, and UVC) irradiation on basic ripening parameters, soluble sugar, organic acid, and carotenoid accumulation during the irradiation period. The primary goal of this work is to determine the most effective light irradiation for postharvest treatment of sweet oranges. The findings not only provide a potential regulation method for fruit quality, but also present important information that can be used in the future for the investigation of the metabolic mechanisms related to these compounds.

## 2. Results

### 2.1. Effects of Light Irradiation on the Total Soluble Solid, Titration Acid, and Citrus Color Index of Postharvest Sweet Oranges

When irradiated by LED and UV light for six days, there was a change in the total soluble solid (TSS) content of the fruit (Figure 1a). Compared with the fruits subjected to dark shade treatment, TSS content was significantly enhanced in fruits treated with red light and UVA, while no significant change was observed in fruits treated with other types of light (*p* < 0.05). The titration acid (TA) content is similar in fruits subjected to all light treatments, other than DS. A significant increase in TA content was observed for the red light, UVA, UVB, and UVC groups compared to the control (DS) (Figure 1b) (*p* < 0.05). The citrus color index (CCI) was remarkably increased by all light treatments in this study (Figure 1c). Compared with the control group, the CCI was strikingly increased in fruits treated with red light, white light, UVA, and UVB (*p* < 0.05). UVA treatment had the most significant positive effect on the CCI.

### 2.2. Effect of Light Irradiation on the Soluble Sugar Content in Postharvest Sweet Oranges

From 0–12 min of retention time, the main three types of soluble sugars detected were fructose, glucose and sucrose, respectively (Appendix A). The total soluble sugar content changed in fruits treated with UV light and LED irradiation for six days. The accumulation of sugar content increased from the harvest value (Day 0) for all light treatments and in the control group (Figure 2). Compared with the control group, only the three kinds of UV light treatments significantly increased the total soluble sugar content (*p* < 0.05) (Figure 2a).

The effects of the lights on different types of soluble sugars were not consistent, which is mainly reflected in the contents of soluble disaccharides and monosaccharides (Figure 2b–d). Similar to the total soluble sugar content, UV light irradiation had a greater positive effect on the fructose content than LED light; compared with the control group, a significant increase in the fructose content was observed in all UV light treatment groups (*p* < 0.05). Among all light treatments, white light presented the smallest effect on the fructose content (Figure 2b). The glucose content significantly increased when fruits were irradiated by UVB and UVC light (*p* < 0.05), while white light led to a decrease in the glucose content (Figure 2c). As for sucrose, white, UVB, and UVC light treatments significantly promoted sucrose accumulation compared with the control group (*p* < 0.05). UVB has the most significant effect on sucrose content, followed by UVC and white light (Figure 2d).

### 2.3. Effect of Light Irradiation on the Organic Acid Content in Postharvest Sweet Oranges

Four types of organic acids detected were citric acid, malic acid, tartaric acid, and quinic acid, respectively (Appendix A). Compared with the control group, except for UVB, other light treatments significantly increased the total organic acid content (*p* < 0.05), with blue light having the largest effect (Figure 3a). Blue and red light irradiation also significantly increased the content of citric acid, the major organic acid in citrus fruit (*p* < 0.05). While white, UVB, and UVC irradiation caused slight increases in the citric acid content, no significant effects on citric acid content were observed for these light treatments compared to the control (Figure 3b). On the other hand, except for UVB, other types of light irradiation caused an increase in the malic acid content relative to the harvest value. However, the inducing effect is only significant between the white light treatment and the control group (*p* < 0.05) (Figure 3c). As for tartaric acid (Figure 3d), white light and the three kinds of UV light, especially UVA, caused significant increases in the tartaric acid content (*p* < 0.05). Although all light treatments presented an inducing effect on quinic acid accumulation in sweet oranges, no significant effect was found, regardless of the light irradiation treatment (*p* < 0.05) (Figure 3e).

### 2.4. Effect of Light Irradiation on Carotenoid Content in Postharvest Sweet Oranges

Carotenoid accumulation in postharvest sweet oranges responded differently to various light irradiation treatments (Figure 4). During the light treatment period, the total carotenoid content and the concentrations of the nine identified carotenoids (isolutein, β-cryptoxanthin, zeaxanthin, lutein, neoxanthin, all-trans-violaxanthin, phytofluene, cis-ζ-carotene, and β-carotene) (Appendix A) effectively increased only in fruits treated with red light and dark shade relative to the harvest value.

The main carotenoids detected in ‘Hamlin’ sweet orange pulp were isolutein, β-cryptoxanthin, zeaxanthin, lutein, neoxanthin, and all-trans-violaxanthin. Red light and dark shade treatments effectively enhanced the contents of these major carotenoids, with concentration increases higher than 1.00 mg kg^−1^ (fresh weight, FW). The concentration changes are lower than 1.00 mg kg^−1^ (FW) for the other five light treatment groups. The results for the six main types of carotenoids showed that the contents of isolutein, neoxanthin, and all-trans-violaxanthin greatly fluctuate for different light treatments (Figure 4b–g). Compared with dark shade treatment, irradiation with blue, white, and the three kinds of UV light caused the total carotenoid, isolutein, neoxanthin, phytofluene, and cis-ζ-carotene contents to significantly drop (*p* < 0.05). The most positive effect was observed for white light treated fruits, followed by the fruits treated with UVC and blue light. It is noticeable that the β-cryptoxanthin content shows high stability across the various light groups.

### 2.5. Principal Component Analysis Based on Soluble Sugar, Organic Acid, and Carotenoid Content in Sweet Oranges Treated with Light Irradiation

To take an overall view of the response of the phytochemicals in sweet orange samples to light treatments, the principal component analysis (PCA) model was developed using unsupervised PCA analysis based on soluble sugars, organic acids, and carotenoids content, respectively. The PCA results demonstrated that the soluble sugars, organic acids, and carotenoids had different responses to the same light sources, after irradiation for six days (Figure 5). In all cases, results can be clearly divided into two or three clusters.

In Figure 5a, for the soluble sugar content the first two principal components account for 98.4% of the total variation, with principal component 1 (PC1) and principal component 2 (PC2) representing 87.9% and 10.5% of the total variation, respectively. The PCA score plot divided the samples into three clusters: the first group contained Day0, the second group contained white light, blue light, and red light, as well as the control group, while the third group contained the three types of UV light. In addition, the total soluble sugar content was distributed in the plot according to the abscissa axis, and the three kinds of soluble sugars have positive correlations with PC1, indicating that the difference in PC1 is mainly controlled by the different types of light treatments. At the same time, all sugars and the three UV light treatments were clustered in the blue group (Group3), while other light treatments were distributed in the green group (Group2). This result is consistent with the differences in sugar accumulation in the various light treated fruits.

As shown in Figure 5b, PC1 (55.6%) and PC2 (30.6%) explained 86.2% of the total variability for the organic acid results. The UVA6, UVC6, WL6, BL6 results, as well as the tartaric acid, malic acid, quinic acid, and total acid were distributed in the red group (Group1), which suggests that UVA6, UVC6, WL6, and BL6 have good correlations with the content of the four organic acids.

As shown in Figure 5c, the first two principal components account for 96.0% of the total variation of the carotenoid content results, of which PC1 accounted for 91.8%. RL6 and DS6 were clustered in the red group (Group1) with the total carotenoid and nine individual carotenoids identified. These results are consistent with the assumption that red light and dark shade treatments induce carotenoid accumulation in fruits.

## 3. Discussion

Light is not only a crucial factor for plant growth and development [18] but has also become an attractive method for maintaining fruit quality after harvest [19]. Thus far, LED irradiation has been extensively used to regulate plant growth and development under protected conditions. In recent years, it has also been used as a postharvest measure to maintain fruit quality. There have been a number of reports about the treatments of UVB and UVC exposure [20,21,22,23], but a global evaluation of fruit quality when exposed to other polychromatic light sources was previously lacking. In the present study, the effects of LED and UV light treatments on the soluble sugar, organic acid, and carotenoid content was evaluated. We found that UV light and red light enhanced the TSS content of sweet oranges, and all light treatments improved the CCI. These results suggest that LED and UV light accelerates the ripening of oranges, similar to what has been observed in bananas and tomatoes [9,24].

Soluble sugar is the key factor determining the taste quality of fleshy fruit; it not only determines the sweetness of fresh fruit, but also directly affects the quality of fruit-processed products [25,26]. For postharvest fruit, rich sugars are also important substrates for many secondary metabolites, which make the plant more resistant to various stresses. When Chinese bayberry was irradiated with blue (470 nm) light at an intensity of 40 μmol m^−2^ s^−1^ for eight days at 10 °C, the gene expression levels of sucrose phosphate synthase (SPS) (associated with sucrose synthesis) and invertase (INV) (responsible for hydrolysis of sucrose into glucose and fructose) significantly increased, resulting in increases in the content levels of sucrose, fructose, and glucose [27]. Similarly, the total soluble sugar content in banana fruit was enhanced by exposure to blue light (464–474 nm), green light (515–525 nm), and red light (617–627 nm) [9]. Likewise, it was found that irradiation with white light (2500 ± 2 lux) at 4 °C for seven days could maintain the contents of soluble sugar and total soluble solid in romaine lettuce [28]. As for UV light, a recent report showed that UVC treatment significantly increased the ratio of total soluble sugars to total organic acid, which was achieved by up-regulation expressions of nicotinamide dinucleotide phosphate-malic enzyme (NADP-ME) and of phosphoenolpyruvate carboxykinase (PEPCK) gene [23]. In this study, all light treatments increased the total soluble sugar content and the individual fructose, sucrose, and glucose contents. This result is consistent with the previous studies. However, it is worth noting that in the present study, we found UV light irradiation to be more effective than treatment with LED light. In particular, UVA had the best positive effect on fructose, UVB had the best positive effect on sucrose, and UVC had the best positive effect on glucose.

Organic acids, mostly citrate and malate, contribute significantly to the citrus fruit flavor and serve as indexes for the senescence status of postharvest fruit [29]. They are metabolized by the tricarboxylic acid (TCA) cycle, resulting in the release of energy in the form of ATP. This process ensures an adequate intracellular ATP supply and extracellular ATP signaling to attenuate stress, delay the aging of horticultural crops, and maintain their quality during postharvest life [30]. Greater accumulation of organic acids was observed in fruits that had been cultivated in plastic greenhouses when ambient light was supplemented with either red LED light or a combination of blue and red LED light [31]. In the present study, the total organic acid content significantly increased after blue, red, white, UVA, and UVC light irradiation but did not significantly change after irradiation with UVB light. Blue and red light significantly increased the content of malic acid, and white light and the three kinds of UV light irradiation remarkably increased the tartaric acid content. These results suggest that the ability of different light treatments to maintain the organic acid content largely depends on the type of organic acid.

Carotenoids, comprising various pigments widely present in nature, not only make horticultural plants more visually rich, but are also an important factor in the human diet. These compounds are rich in sweet oranges [32,33]. Up to now, that carotenoid accumulation depends on the type of light, was found in some crops and postharvest fruits. A previous study showed that treatment with 16–33% blue light had a positive effect on the carotenoid content in microgreens [34]. Further, carotenoid biosynthesis was up-regulated in sprouts treated under white light conditions [35], and an appropriate UVB dose induced the enrichment of carotenoids in germinated corn kernels [36]. In tomatoes, postharvest UVB (1 h, 6.08 kJ/m^2^ d) treatment increased the concentration of carotenoids [24]. Green tomatoes were exposed for 30 min to UV radiation, continuous red light, or a combination of both for up to 20 days. The exposure to red light alone or that in combination with UV raised the concentrations of lycopene and β-carotene [37]. Light red tomatoes were exposed to different doses of UVC irradiation (1.0, 3.0, and 12.2 kJ m^−2^) for 1, 3, or 12 h. When the tomatoes were stored for two days at room temperature, the lycopene content was found to have increased by 14% with respect to the control samples, while the β-carotene content decreased. Cis-isomers from lycopene also increased when the tomatoes were exposed to light for more than 3 h [38]. Compared with the control group, the concentration of lycopene in tomato exocarp was significantly increased after four days and was dramatically enhanced by the UVC or red light treatments; however, the concentration of β-carotene was not affected by UVC or red light treatments and decreased following sunlight treatment during 21 days of storage [39]. In the flavedo of the Satsuma mandarin, β-cryptoxanthin, all-trans-violaxanthin, 9-cis-violaxanthin, and lutein increased simultaneously along with the total carotenoid accumulation when irradiated with red LED light (660 nm), which contributed to increases in the expression levels of *CitPSY*, *CitPDS*, *CitZDS*, *CitCRTISO*, *CitLCYb1*, *CitLCYb2*, *CitLCYe*, *CitHYb*, and *CitZEP* in the carotenoid biosynthesis pathway [40,41]. In the present study, we found that nine carotenoids were identified from the ‘Hamlin’ sweet orange, and these carotenoids showed a consistent response to light irradiation, their concentrations in the citrus fruit were efficiently increased by postharvest red light irradiation, but presented a significant decrease in other types of light treatment fruits. These above findings support the conclusion that postharvest red light improves carotenoid accumulation in citrus fruits.

Also, previous studies have shown that the carotenoid accumulation depends on the light intensity. In tea, the contents of chlorophyll and β-carotene significantly increased following shading treatments [42]. Changing the light intensity using color shade nets affected the biosynthesis of lycopene and β-carotene in tomatoes, with a significantly higher lycopene content observed in greenhouse tomatoes integrated with red shade netting technologies than in field-grown tomatoes [43]. In bananas, blue, green, and red light sped up peel de-greening and flesh softening, increased ethylene production and the respiration rate, and enhanced the accumulation of ascorbic acid, total phenols, and total sugars in the fruit flesh [9]. In pears irradiated with UVB (280–315 nm) and/or fluorescent lamps for 10 days after harvest, significant increases in carotenoid and flavonoid concentrations in the peel, slight increases in soluble solids and organic acid concentrations, and a significant increase in the total sugar content in the flesh were reported [44]. In a recent study, we found that the accumulation of flavonoids and limonoids in the segments of ‘Newhall’ navel oranges were largely enhanced [45]. These studies indicate that postharvest irradiation can permeate through the peel of the whole fruit to affect the inner flesh. Irradiation of the peel is a good nondestructive method that could be widely used to regulate fruit quality. However, the intensity of light may be different for each part of the fruit. The result of our study showed that the treatment of dark shade treatment decreased light intensity and efficiently induced the accumulation of all carotenoids. In regard to this, further work is required to confirm the intensity and quality of light reaching the flesh.

In addition, the combination of polychromatic lights may have a different effect on carotenoid accumulation in plants. In microgreens, treatment with 16%–33% blue light had a positive effect on the photosynthetic and carotenoid pigments, and changes in metabolite quantities were influenced by light treatment and depended on the species [34]. In a system of five high-power, solid-state lighting modules with standard 447, 638, 665, and 731 nm wavelength LEDs, the concentrations of various carotenoids in red pak choi and tatsoi were higher under illumination at 330–440 μmol m^−2^ s^−1^, and at 110–220 μmol m^−2^ s^−1^ for mustard. All supplemental wavelengths increased the total carotenoid content in mustard but decreased it in red pak choi, and the carotenoid content increased in tatsoi under supplemental yellow light [46]. Therefore, the use of different light combinations may be a good direction for the regulation of fruit quality.

Light, largely in terms of signaling, is involved in the regulation of carotenoid accumulation. Orange head Chinese cabbage results from a defect in carotenoid isomerase (CRTISO). When the cabbage is exposed to light, light-induced isomerization leads to the production of lycopene, which indicates that CRTISO activity can be partially replaced by light [47]. In tomatoes, the suppression of the light signaling gene *HY5* leads to decreased carotenoid levels, and the down-regulation of *COP1LIKE* causes increased carotenoid accumulation, revealing that *HY5* and *COP1LIKE* play positive and negative roles in controlling fruit pigment accumulation, respectively [48]. Light-triggered degradation of phytochrome-interacting factor (PIF) after interaction with photoactivated phytochromes during de-etiolation results in a rapid depression of *PSY* gene expression and a burst in the production of carotenoids in coordination with chlorophyll biosynthesis and chloroplast development, allowing an optimal transition to the photosynthetic metabolism. This suggests a role for PIF1 and other PIFs in transducing light signals to regulate *PSY* gene expression and carotenoid accumulation during daily cycles of light and dark in mature plants [49]. On the basis of these results, for developmental plants, carotenoid accumulation is mainly determined by photosynthesis and metabolite conversion, however accumulation in postharvest plants only depends on metabolite conversion. The specific regulation light signaling mechanism still needs to be investigated. Even so, combined with the existing related reports, in this study, we concluded that red light and dark shade treatments effectively enhance the carotenoid content in postharvest fruit as a result of changes in the gene expression of carotenogenic genes. The results presented might provide new strategies to enhance the commercial and nutritional value of citrus fruits using appropriate light treatments.

## 4. Materials and Methods

### 4.1. Fruit Materials

The ‘Hamlin’ sweet orange (*Citrus sinensis* (L.) Osbeck) was collected from the National Citrus Germplasm Repository of the Citrus Research Institute at the Chinese Academy of Agricultural Sciences in Chongqing, China. The fruits were picked randomly from different positions of the orchard according to their external color and uniform size at the commercial maturity stage. When transported to the laboratory, fruits without mechanical damage and pests were grouped into eight groups; 60 fruits were included in each group, with 20 fruits for each biological replicate. All analyses were performed in triplicate.

### 4.2. Postharvest Treatments

Fruits were irradiated in an RXZ-300D intelligent growth cabinet with adjustable light irradiation parameters (Southeast Instrument Company, Ningbo, China), and three kinds of UV lamps (TL-D36/16, Philips Company, Amsterdam, The Netherland). And three types of LED lamps (RDN-500B-C, Southeast Instrument Company, Ningbo, China) were installed. Fruits were irradiated with red light (660 nm, 150 μmol m^−2^ s^−1^), blue light (470 nm, 200 μmol m^−2^ s^−1^), white light (100 μmol m^−2^ s^−1^), UVA (315–400 nm, 100 μmol m^−2^ s^−1^), UVB (270–315 nm, 100 μmol m^−2^ s^−1^), or UVC (100–280 nm, 100 μmol m^−2^ s^−1^) for six days, and dark shade treatment was used as the control. LED intensities were adjusted by the incubator. UV lamps had fixed light intensities, and all light intensities were confirmed by a spectrum analyzer (HR-350, Taiwan, China). The different light doses were obtained by exposure at a fixed distance, and fruits were placed 20 cm below the lamps. During the treatment period, fruits were rotated vertically by 180° every eight hours to ensure that all parts of the fruit were exposed to the same level of light irradiation. The relative humidity (RH) was maintained at 90%–95%, and the temperature was set at 20 °C. For each group, after determining the basic ripening indexes, other edible section segments were sliced into small cubes and frozen using liquid nitrogen, followed by storage at −80 °C until analysis.

### 4.3. Determination of the Basic Ripening Parameters

Color changes were determined using the Hunter Associates Laboratory Scanner (Hunter Associates Laboratory, Inc., Reston, VA, USA). The Commission Internationale de I’Eclairage (CIE) L*a*b*color scale was adopted, and the data were expressed as L*, a*, b*, C*, and H. Next, the shaded and exposed sides of each fruit were measured under the light source of the irradiation standard D65. The CCI was calculated according to the formula CCI = 1000 × a*/(L* × b*). Five fruits were used as one biological replicate, and three replicates were used for each sample.

After determining the color value, the segments were pressed to obtain their juices. The TSS content was measured using a mixture of the juices and a digital hand-held refractometer (Atago PR-101R, Atago, Japan). The juices were diluted 100 times with pure water to determine the TA content using the refractometer, and the content was expressed in %.

### 4.4. Determination of the Soluble Sugar and Organic Acid Contents

The soluble sugar and organic acid contents were extracted using our previously described method [25]. Three grams of frozen segment cubes were ground into a uniform powder, 5 mL of 80% ethanol was added, and then the mixture was homogenized by a vortex. Next, the solution was incubated in water at 35 °C for 20 min to extract the soluble sugars and organic acids. After centrifugation at 10,733 g for 15 min at 4 °C, the supernatant was transferred to a 15 mL centrifuge tube. All extraction procedures were repeated three times. Next, the supernatant was removed and mixed with 80% ethanol to give a 15 mL final solution. Four milliliters of supernatant were removed from the final solution and centrifuged at 10,733 g for 5 min at 4 °C. Three milliliters of the supernatant was removed and then dried using nitrogen. Additionally, the dry residue was dissolved in 1.5 mL of double-distilled water. Finally, the solution was passed through an organic membrane filter with a pore diameter of 0.22 μm.

Soluble sugars and organic acids were analyzed by high-performance liquid chromatography (HPLC), as described previously with some modifications [25]. The chromatographic separation of soluble sugars was conducted on a 0.5 μm NH_2_ column (4.6 mm × 250 mm, Shimadzu, Japan), and were detected with a 2424 evaporative light scattering detector (ELSD) (Waters Beckman Coulter Inc., Brea, CA, USA). Organic acids were separated on a 0.5 μm C18 column (250 mm × 4.6 mm, Agilent, USA) and detected by a 2998 photodiode array detector (PDA) (Waters Beckman Coulter Inc., Brea, CA, USA).

The soluble sugar and organic acid contents were identified according to the retention time of the sample and standards. The content was quantified using the standard curve method, and the final result was expressed as g kg^−1^ on a fresh weight (FW) basis. Three biological replicates were used for each sample. Soluble sugar (fructose, glucose, and sucrose) and organic acid (citric acid, malic acid, tartaric acid, and quinic acid) standards were all obtained from Shanghai Sangon Biological Reagent Company (Shanghai, China).

### 4.5. Determination of Carotenoid Content

The extraction of carotenoids was carried out according to our previously described method [25]. Ten grams of frozen segments were ground into a powder and placed into a 50 mL centrifuge tube, and then 50 mL of extracting solution (hexane/acetone/ethanol, 50:25:25, *v*/*v*/*v*) was added. After homogenization, the mixture was centrifuged at 4535 g and 5 °C for 4 min. Next, the mixture was allowed to stand for more than 30 min, and the colored hexane top layer was recovered and transferred to a 25 mL volumetric flask. Thereafter, the sample was dried with nitrogen and then dissolved in 2 mL of methyl tert-butyl ether (MTBE). Then, the solution was transferred to 2 mL of 10% methylated potassium hydroxide solution and shaken in a shaker overnight at 25 °C. The next day, 5 mL of water was added and allowed to stand in a separatory funnel. The solution was rinsed twice with 5 mL of water and once with 2 mL of 0.1% butylhydroxytoluene (BHT)/MTBE. The MTBE layer was filtered through sodium sulfate and dissolved in 2 mL of methanol/acetone (2:1, *v*/*v*) for detection.

Thereafter, 1.5 mL of solution was injected into a HPLC system (Waters, Milford, MA, USA) using a C30 column (250 mm × 4.6 mm, 5 μm) (YMC Inc., Wilmington, NC, USA). Carotenoids were identified by comparing the standard retention time and UV light-emitting diode spectral peaks. The HPLC mobile phase used to identify the carotenoids comprised MTBE (eluent A), methanol (eluent B) and an aqueous phase (eluent C), and elution occurred in 10% and 50% methanol solutions. The elution program was set as follows: 0–12 min, 95% B/5% C; 12–25 min, 5% A/95% B; 25–40 min, 11% A/89% B; 40–60 min, 25% A/75% B; 60–66 min, 50% A/50% B; 66–68 min, 95% B/5% C, and back to the initial conditions for equilibration.

The standard curve method was used to quantify the carotenoid content, expressed as mg kg^−1^ (FW). Three biological replicates were used for each sample point. Carotenoid standards, including isolutein, β-cryptoxanthin, zeaxanthin, lutein, neoxanthin, all-trans-violaxanthin, phytofluene, cis-ζ-carotene, and β-carotene, were obtained from Sigma (St. Louis, MO, USA).

### 4.6. Statistical Analysis

Three biological replicates were used for all experiments, and the results are expressed as the mean and standard error of biological triplicates. SPSS 10.01 (SPSS Inc., Chicago, IL, USA) was applied to perform the statistical analysis. Origin Pro 7.5 G (Microcal Software, Inc., Northampton, MA, USA) was used for drawing the histograms. Principal component analysis (PCA) was performed by Rstudio (Version 1.1.463) using the mean of three biological replicates of each parameter. Statistically significant differences were analyzed by Fisher’s protected least significant difference (LSD) test with *p* = 0.05 representing the significance.

## 5. Conclusions

In this study, postharvest LED and UV light treatments accelerated the ripening of sweet oranges. The accumulation of soluble sugars, organic acids, and carotenoids in ‘Hamlin’ sweet oranges occurred differently under various light irradiation conditions. UV light strikingly induced the accumulation of soluble sugars. Red light and blue light treatment promoted notable increases in the total organic acid and citric acid contents. Red light and dark shade effectively promoted carotenoid accumulation in sweet orange. Further research is needed to investigate the molecular mechanisms responsible for the specific accumulation of different compounds.

## Figures and Tables

**Figure 1 molecules-24-03440-f001:**
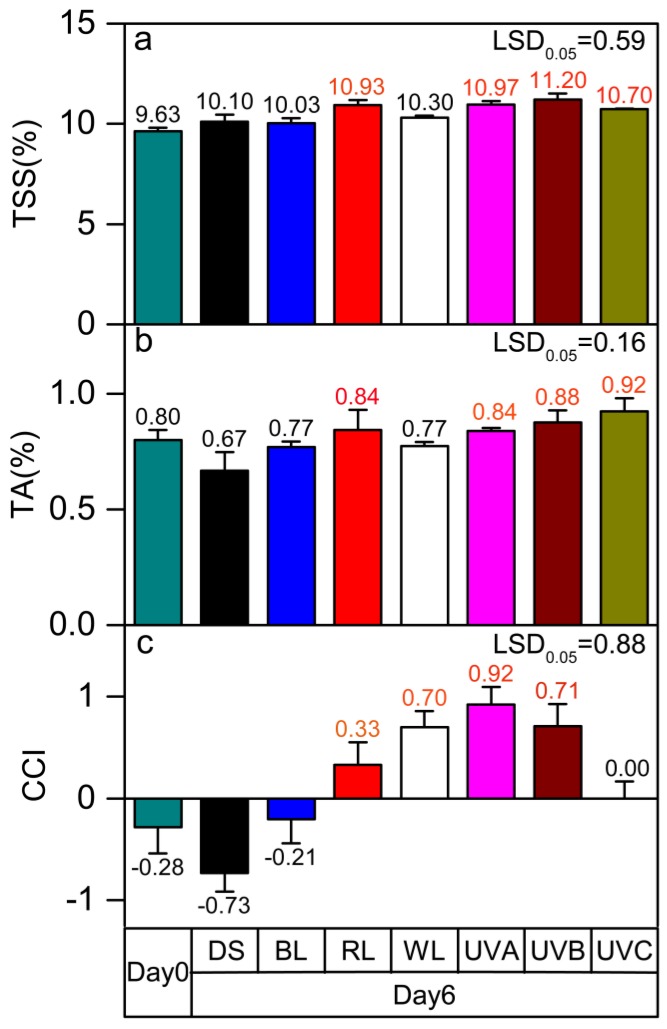
Effects of different light irradiation treatments on the (**a**) total soluble solids (TSS) content, (**b**) titration acid (TA) content, and (**c**) citrus color index (CCI) of postharvest fruit. TSS—total soluble solid; TA—titratable acidity; CCI—citrus color index; DS—dark shade; WL—white light; BL—blue light; RL—red light; UVA—ultraviolet A; UVB—ultraviolet B; UVC—ultraviolet C. Other than the harvest day measurements (Day 0) all other cases were analyzed after six days. The dark shade treatment is the control group. LSD represents the least significant difference (*p* < 0.05), and each red value indicates a significant difference at this level. Each level and error bar represents the mean and standard error of three biological replicates (*n* = 3).

**Figure 2 molecules-24-03440-f002:**
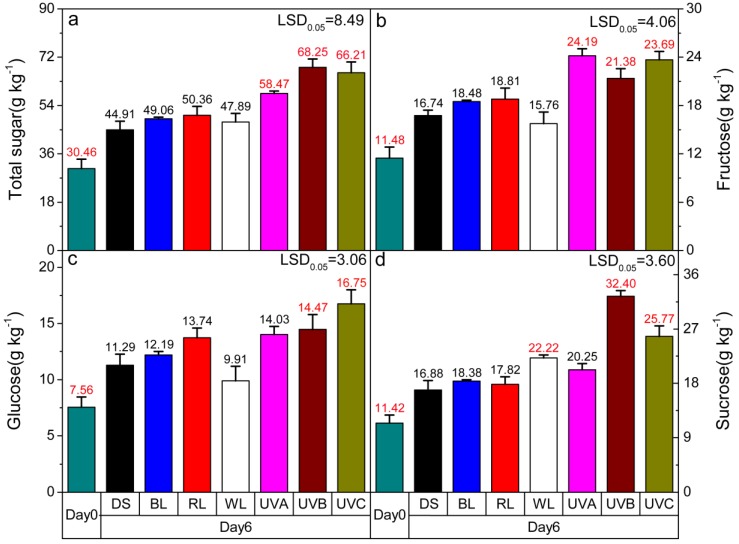
Effects of irradiation with different types of light on the (**a**) total soluble content, (**b**) fructose content, (**c**) glucose content, and (**d**) sucrose content of postharvest fruit. DS—dark shade; WL—white light; BL—blue light; RL—red light; UVA—ultraviolet A; UVB—ultraviolet B; UVC—ultraviolet C. Other than the harvest day measurements (Day 0) all other cases were analyzed after six days. The dark shade treatment is the control group. LSD represents the least significant difference (*p* < 0.05), and each red value indicates a significant difference at this level. Each level and error bar represents the mean and standard error of three biological replicates (*n* = 3).

**Figure 3 molecules-24-03440-f003:**
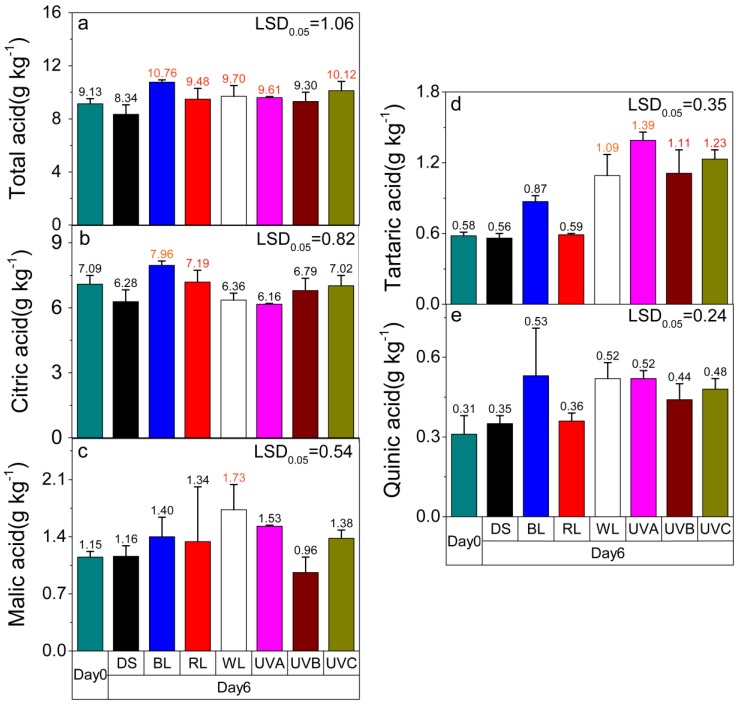
Effects of different light irradiation treatments on the (**a**) total acid content, (**b**) citric acid content, (**c**) malic acid content, (**d**) tartaric acid content and (**e**) quinic acid content in postharvest sweet orange fruit. DS—dark shade; WL—white light; BL—blue light; RL—red light; UVA—ultraviolet A; UVB—ultraviolet B; UVC—ultraviolet C. Other than the harvest day measurements (Day 0) all other cases were analyzed after six days. The dark shade treatment is the control group. LSD represents the least significant difference (*p* < 0.05), and each red value indicates a significant difference at this level. Each level and error bar represents the mean and standard error of three biological replicates (*n* = 3).

**Figure 4 molecules-24-03440-f004:**
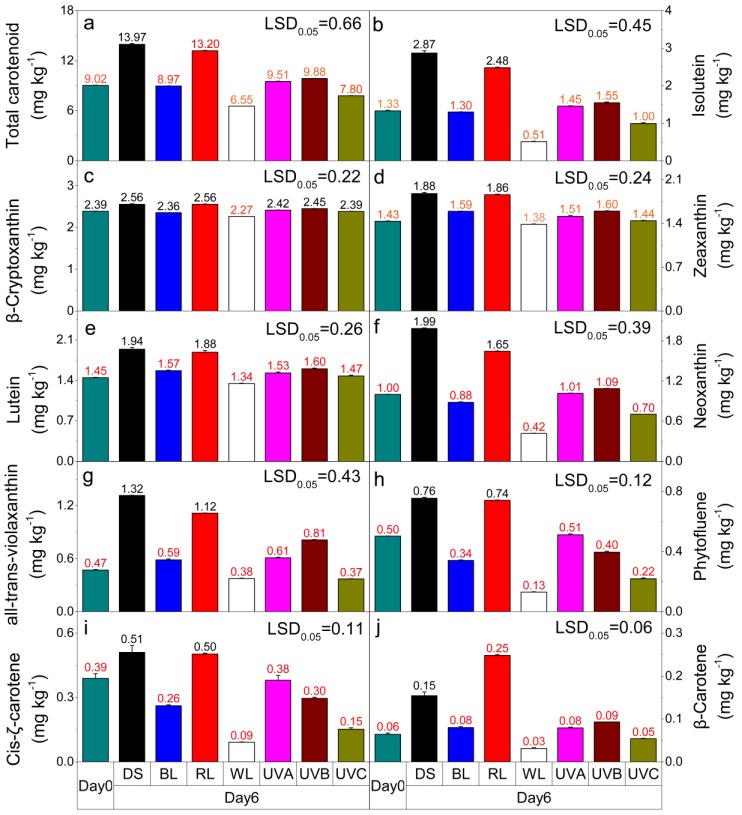
Effects of different light irradiation treatments on the (**a**) total carotenoid content, (**b**) isolutein content, (**c**) β-cryptoxanthin content, (**d**) zeaxanthin content, (**e**) lutein content, (**f**) neoxanthin content, (**g**) all-trans-violaxanthin content, (**h**) phytofluene content, (**i**) cis-ζ-carotene content, and (**j**) β-carotene content in postharvest sweet orange fruit. DS—dark shade; WL—white light; BL—blue light; RL—red light; UVA—ultraviolet A; UVB—ultraviolet B; UVC—ultraviolet C. Other than the harvest day measurements (Day 0) all other cases were analyzed after six days. The dark shade treatment is the control group. LSD represents the least significant difference (*p* < 0.05), and each red value indicates a significant difference at this level. Each level and error bar represents the mean and standard error of three biological replicates (*n* = 3).

**Figure 5 molecules-24-03440-f005:**
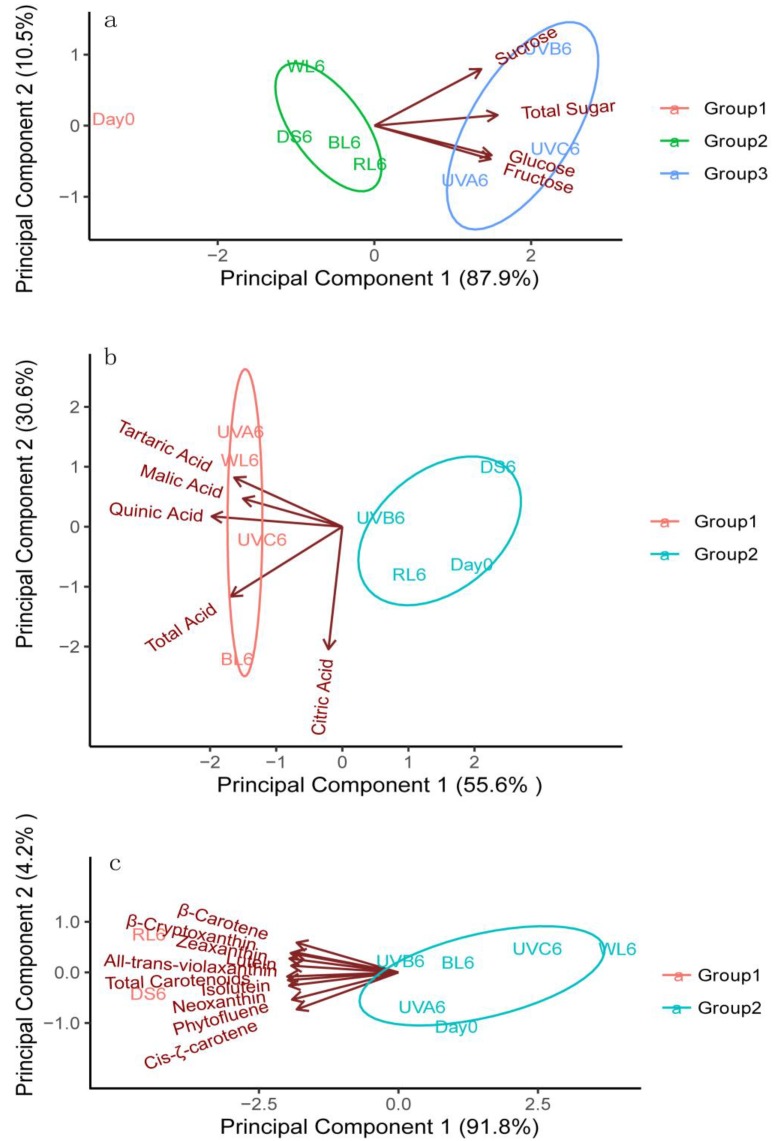
Principal component analysis (PCA) models for the (**a**) soluble sugar, (**b**) organic acid, and (**c**) carotenoid content in sweet oranges after various treatments. DS—dark shade; WL—white light; BL—blue light; RL—red light; UVA—ultraviolet A; UVB—ultraviolet B; UVC—ultraviolet C. Other than the harvest day measurements (Day 0) all other cases were analyzed after six days. The dark shade treatment is the control group.

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
