# Peer review of "Effect of Light-Emitting Diodes and Ultraviolet Irradiation on the Soluble Sugar, Organic Acid, and Carotenoid Content of Postharvest Sweet Oranges (Citrus sinensis (L.) Osbeck)"

_molecules, 2019, doi:10.3390/molecules24193440_

Round 1

Reviewer 1 Report

In this study, the effects of different LED and UV lights on the postharvest Sweet orange were investigated. The authors concluded that different light irradiation might be an effective way to improve the fruit quality after harvest. However, the results presented in this study did not provide enough proof to support the conclusion.

In this study, the authors investigated the effects of different lights on the nutritional contents in the segments. Why did they choose these lights? Can the blue, red and white light permit the peels? In Fig. 1, the differences in the TSS% and TA% were not significant among different treatments. In addition, I think that the description in the figure legends was not correct, as the results did show the contents of TSS and TA. The results of carotenoids accumulation was quite difficult to understand. In Fig. 4, which one was used as the control? I think that different lights did not significantly affect the carotenoid accumulation.

Author Response

Comments and Suggestions for Authors:

In this study, the effects of different LED and UV lights on the postharvest Sweet orange were investigated. The authors concluded that different light irradiation might be an effective way to improve the fruit quality after harvest. However, the results presented in this study did not provide enough proof to support the conclusion.

Response: Thank you for pointing it out. In fact, in the present study we only compared the effect of different light on phytochemicals accumulation in fruits, but the related mechanism is not fully understood. According to the suggestion, to strengthen our conclusion, some related information was added in “Discussion”, and possible mechanism was discussed in the revised manuscript, see pages 9-10, lines 228-294 without tracked changes.

Point 1: In this study, the authors investigated the effects of different lights on the nutritional contents in the segments. Why did they choose these lights?

Response 1: Yes, in the study, we investigated the effects of different lights on the nutritional contents in the fruit. At present, a large number of studies, including white light, red light and blue light in visible light, and ultraviolet light (UVB and UVC), are widely used to treat fruit and monitor the changes of quality, but almost all studies only used one of the light or two kinds of irradiation. In the study, we aimed to conduct a comprehensive comparison of several common lights, and obtain the optimal irradiation that is suitable for practice. In addition, the intensity for each light were confirmed based on many previous studies that proved only some specific intensity for one kind of light is efficient for postharvest treatment (Ma et al. Journal of Agricultural and Food Chemistry 2012 60 (1), 197-201; Harbaum-Piayda et al. (2010). Postharvest Biology and Technology 56(3): 202-208; Ma, G., et al. (2015). Postharvest Biology and Technology 99(Supplement C): 99-104; Atta, M., et al. (2013). Bioresource Technology 148: 373-378.). Based on the comment, the related discussion about light quality, intensity and combination of different light was also added in the Discussion section, page 9, lines 228-273.

Point 2: Can the blue, red and white light permit the peels?

Response 2: Thanks for your pointing it out. We are deeply sorry for that we are unable to provide the direct evidence whether the light can permeate through the peels and affect phytochemicals accumulation in the segment, but we have found a related report that may indirectly prove this point. Similar with banana, the blue, green and red light speed up the peel de-greening and flesh softening, and increased ethylene production and respiration rate in bananas. Furthermore, the accumulations of ascorbic acid, total phenols, and total sugars in banana fruit flesh were enhanced by LED light exposure, the results suggested that blue, green and red light can penetrate banana peel (reference 9, Huang et al., Journal of the Science of Food and Agriculture, 2018, 98, 5486-5493.). In addition, Irradiation of the entire pear fruits with UVB and/or fluorescent lamps for 10 days after harvest can not only promote the accumulation of various carotenoids and flavonoids in the peel (reference 45, Figure 3, Sun, Y et al. Postharvest Biology and Technology, 2014, 91, 64-71), but also lead to slight increase of soluble solids and organic acids concentration and significant increase of total sugars in the flesh (reference 45, Figure 4, Sun, Y et al. Postharvest Biology and Technology, 2014, 91, 64-71). The two studies indicated that postharvest irradiation can permeate through the peel of the whole fruit to affect the inner flesh.

In the present study, we do show that different light treatments lead to changes in phytochemical of citrus fruit edible parts, which is hard to explain if light hasn’t affected the flesh. According to the suggestion, we have added the discussion in the revised manuscript, see page 9, line 255-264.

Point 3: In Fig. 1, the differences in the TSS% and TA% were not significant among different treatments. In addition, I think that the description in the figure legends was not correct, as the results did show the contents of TSS and TA.

Response 3: In the study, the statistically significant differences were analysed by Fisher's protected least significant difference (LSD) test at a P = 0.05 level. Yes, the TA were not significant among different treatments at 0.05. The related description was corrected, see page 2, lines 73-75. In the revised Fig.1, the original values were presented on all columns, which is convenient to compared with the control. The error in the legend was corrected. Please see page 3, lines 82-85.

Point 4: The results of carotenoids accumulation was quite difficult to understand.

Response 4: Sorry for not expressing clearly. In the study, at harvest (day 0) is the control, So it is very clear that the contents of the total carotenoid and nine carotenoids identified were significantly increased only in fruits treated with red light and dark shade (P < 0.05), these two treatments presented the best inducing effect on carotenoid accumulation. See page 5, lines 128-146.

Point 5: In Fig. 4, which one was used as the control? I think that different lights did not significantly affect the carotenoid accumulation.

Response 5: I am sorry for our unclear description. In our study, the fruits after harvest at day0 is the control. According to the suggestion, the related description was revised, see page 2, line 70. So in this study, it is clear that the carotenoid content only in dark shade and red light irradiation group were induced significantly compared with the control group. According to the suggestions, the information about control was added in all Figure legends in the revised manuscript. In addition, the information was also added in the related description the revised manuscript.

Reviewer 2 Report

The author is invited to consider the comments and suggestions in the annotated document attached to this review. Special consideration should be given to the description of light treatments. At this point there are serious lacks that will prevent the experiment from being repeated  by other research teams. Among these lacks are:

What instrument was used for the measurement of light intensity? What was the distance between the fruit and the light sources? Were the fruit rotated during treatment? Were the fruit continuously subjected to these treatments over the entire 144 hours of the experiment, with no period of darkness?

The discussion section is also very weak and needs to be strengthened.

Overall, the level of English must be  thoroughly improved by professional of the English language.

Author Response

Comments and Suggestions for Authors:

The author is invited to consider the comments and suggestions in the annotated document attached to this review. Special consideration should be given to the description of light treatments. At this point there are serious lacks that will prevent the experiment from being repeated by other research teams. Among these lacks are:

Point 1: What instrument was used for the measurement of light intensity?

Response 1: We are sorry for unclear explanation. In order to ensure the accuracy of the light intensities, LEDs intensities were adjusted by the incubator, UV lamps have fixed light intensities, and all light intensities were confirmed by the spectrum analyzer (HR-350, Taiwan, China). See page 10, lines 308-313 without tracked changes.

Point 2: What was the distance between the fruit and the light sources? Were the fruit rotated during treatment? Were the fruit continuously subjected to these treatments over the entire 144 hours of the experiment, with no period of darkness?

Response 2: Thank you for pointing it out. The different types of light doses were obtained by the exposure at a fixed distance, and fruit were placed 20 cm below the lamps. During treatment period, every eight hours, fruits were rotated vertically by 180° to ensure all fruits exposed to the same light irradiation. See pages 10, lines 313-316.

Yes, in the study, citrus fruits were continuously subjected to light irradiation treatments over the entire 144 hours of the experiment, and there was no period of darkness. According to the existing literature, and the continuous treatment was used in previous study (Ma G , Zhang L , Kato M , et al. Journal of Agricultural and Food Chemistry, 2012, 60(1):197-201.).

Point 3: The discussion section is also very weak and needs to be strengthened.

Response 3: Thank you for pointing this out. We have strengthened the discussion section in the revised manuscript, please see discussion section in the revised manuscript, pages 8-10, lines 179-294.

Point 4: Overall, the level of English must be thoroughly improved by professional of the English language.

Response 4: The English language was improved by using MDPI language service system. See the revised manuscript.

The following points are from "molecules-572129-review.pdf":

Abstract

Point 1: Line 14: "for six days after harvest", Were the treatments applied continuously over the 6 days? It would be surprising that the fruit are able to withstand such long-term light treatments.

Response 1: Thank you for your pointing it out. In fact, when designing the study, we referred the previous study in citrus (Ma et al. (2012). Journal of Agricultural and Food Chemistry 60 (1):197-201.), and the six-day light treated was also used in the existing citrus work (Choi, H. G., et al. (2015). Scientia Horticulturae 189: 22-31.). In addition, before conducted the formal treatment, we make a preliminary experiment to confirm the security treatments time, our results shown that the continuous irradiation for 6 days have not effect on fruit colour, flavour and firmness, and also not lead to fruit decay.

Point 2: Line 23: "These light irradiation is a", Please make appropriate corrections.

Response 2: It is corrected as “Appropriate light irradiation is a potentially”in the revised manuscript, see page 1, line 25-26.

Introduction

Point 3: Line 43: I suggest revising "single-color light" to "monochromatic light".

Response 3: It is corrected according to the suggestion. see page 2, line 45.

Point 4: Line 44: I suggest revising "multiple-color" to "polychromatic light".

Response 4: It is corrected according to the suggestion, see pages 2, lines 46.

Point 5: Line 45: I suggest "…especially both LED and UV light" without the word "both".

Response 5: the word "both" was removed from the sentence, see page 2, line 47.

Point 6: Lines 48-49: "but understanding the role of", This sentence is misleading. The present study does not disclose new aspect regarding the physiological comprehension of the effect of UV and LED on postharvest fruit quality.

Response 6: Thank you for pointing it out. The sentence was corrected as “but understanding the role of postharvest UV and LED irradiation in the accumulation of fruit quality-related metabolites is very limited.” according to the suggestion, see page 2, lines 50-52.

Point 7: Line 57: "while", Please verify.

Response 7: It was corrected as “white”, see page 2, line 60.

Point 8: Lines 59-60: "and to ultimately determine the most effective light irradiation for postharvest treatment", Start a new sentence.

Response 8: The sentence was corrected as “and ultimately determine the most effective light irradiation for postharvest treatment.”, see page 2, lines 62.

Result

Point 9: Line 66: "The number after the light", Please revise to improve English quality.

Response 9: The wrong expression was corrected as “The number after the day represents the number of days after harvest”, see page 3, lines 84-85. Similar errors in other figure legends were also corrected.

Point 10: Line 74: I suggest revising "irradiated by three kinds of" to "irradiation with the three types of".

Response 10: It is rewritten as “Among them, compared with the harvest (day 0), TSS contents were significantly enhanced in fruits treated with UVA, UVB, UVC, white light, and red light, while no significant change was observed in fruits treated with blue light or the control (P < 0.05).” , see page 2, lines 70-73.

Point 11: Line 77: I suggest "…treated with UV light and red light irradiation were…"without the word "irradiation".

Response 11: It is rewritten as “no significance was observed between the control group and any light treatment group”, see page 2, line 75.

Point 12: Line 81: I suggest revising "irradiations" to "light treatments".

Response 12: According to the suggestion, "irradiations" was revised as "light treatments" (page 2, line 76), similar expression was also revised.

Point 13: Line 81: I suggest revising "the" to "this".

Response 13: It is corrected, see page 2, line 76.

Point 14: Line 93: "segments of postharvest", This is ambiguous.

Response 14: The expression was corrected. See all figure legends.

Point 15: Line 95: "The number after the light represents", Please revise to improve English quality.

Response 15: It is corrected, see page 3, lines 84-85. Similar expression was also revised in other figure legends.

Point 16: Line 100: "embodied", This not the correct word to use here.

Response 16: It is corrected as “In particular, the fructose content significantly increased in the UVA and UVC light treatment groups”, see page 3, lines 98-99.

Point 17: Line 100: "... when fruits irradiated by UV light", adding the word "were" to the word "fruits".

Response 17: It is corrected, see page 3, lines 100.

Point 18: Line 118: I suggest revising "and" to ",".

Response 18: It is corrected as “UV light, red light”, see page 4, lines 112.

Point 19: Line 119: I suggest revising "did not have" to "had no".

Response 19: It is corrected as “white light had no significant”, see page 4, lines 112.

Point 20: Line 131: "all-z-violaxanthin", verify if it should not be "all-trans-violaxanthin".

Response 20: It is corrected, see page 5, lines 130.

Point 21: Line 132: "significantly induced", total carotenoid as depicted in Fig 4a was increased only in NL and RL treated fruits. Please revise. The author should consider presenting the data in tables. It will be easier for the readers to compare the measured values.

Response 21: We are sorry for our wrong description. In our study, the Day0 is the control, Yes, you are right, the carotenoids were significantly only induced by dark shade and red light treatment. See page 5, 131-132.

According to the suggestion, to be convenient for comparison, the values for all column was added in all figures and “the day0 is the control” was added in all figure legends. See all revised manuscript.

Point 22: Lines 142-143: "red light irradiation can effectively maintain their contents, and their contents of the above carotenoids in the red light", This is ambiguous. Please revise to improve English level.

Response 22: It is corrected as “red light treatment effectively enhanced the contents of these major carotenoids,”, see page 5, lines 135-136.

Point 23: Lines 144: "1.00 mg kg-1 (fw), while were lower than 1.00 mg kg-1 (fw)", It is not clear if these figures significantly different.

Response 23: It is corrected, see page 5, lines 136-138.

Point 24: Lines 164: "on seven variables", Please verify.

Response 24: The sentence was deleted.

Point 25: Lines 170-171: "indicating that the difference of PC1 was mainly contributed by soluble sugar, and UV light had the most influential on soluble sugars accumulation in sweet orange", Revise to improve clarity.

Response 25: the sentence was revised as “indicating that the difference of PC1 was mainly contributed by different types of light treatment, and UV light had the most influence on soluble sugar accumulation in sweet oranges”, see page 7, lines 168-169.

Point 26: Line 172: "the seven variables", Please verify and improve clarity. Verify quinic acid spelling in the figure.

Response 26: The related information was deleted. The “qinic acid” was corrected as “quinic acid”.

Point 27: Lines 176-177: "Due to the low contribution rate of PC1 and PC2, it indicated

that the cause of organic acids changes by light exposure need further study", Not clear. PC1 and PC2 explain 86.2% of the total variation, that's significant.

Response 27: The sentence was rewritten as “In Fig. 5b, PC1 (55.6%) and PC2 (30.6%) could explain 86.2% of the total variable. UVA6, UVC6, WL6, BL6, tartaric acid, malic acid, quinic acid, and total acid were distributed in the red group 1, which suggests that UVA6, UVC6, WL6, and BL6 had a good correlation with the four organic acids.”, see page 7-8, lines 170-173.

Point 28: Line 179: suggesting revising "variables" to "variation".

Response 28: "variables" was revised as "variation". See page 8, 174.

Point 29: Line 184: "red light", Comment should also be made about the position of NL6 as it relate to PC1 and the different carotenoid vectors.

Response 29: The sentence was deleted. See page 8, 173-177.

Discussion

Point 30: Lines 196-197: "…large progress has been made only in UVB treatment", This is not exact. Work have been also conducted with UV-C. For an accurate report on the state of the art, the author could search on Scopus using the keywords "UV-C and Postharvest"

Response 30: According to the suggestion, the sentence was revised as “A good amount of progress has been made in UVB and UVC treatments”, see page 8, lines 182-183.

Point 31: Line 199: "after treatments", Does the author mean "during and after storage"?

Response 31: The sentence was deleted. see page 8, lines 184.

Point 32: Lines 199-200: "LEDs and UV light simultaneously", This suggests that LEDs and UV light were applied to the same samples at the same time?

Response 32: The sentence was revised as “LEDs as well as UV light”, see page 8, lines 184-185.

Point 33: Line 211: I suggest revising "can induce the increase of" to "increased".

Response 33: It was corrected as “all light treatments increased total soluble sugar”, page 8, lines 204.

Point 34: Line 212: I suggest revising "above, however" to "above. However".

Response 34: It was corrected as “with the previous studies. However”, see page 8, lines 205.

Point 35: Line 213: I suggest revising "among which in our research" to "in our study".

Response 35: It was revised as "in the present study", see page 8, lines 205.

Point 36: Line 213: I suggest revising "better significant" to "more effective".

Response 36: It was revised as " more effective ", see page 8, lines 206.

Point 37: Line 220: "friendly", It is not appropriate to use this word here.

Response 37: "friendly" was deleted, see page 8, lines 212.

Point 38: Lines 224-225: "Therefore, we speculate that the maintain ability for light irradiation largely depended on the organic acids accumulation in citrus fruits." Very poorly expressed. Please work with a English language professional to improve the quality of the written text.

Response 38: It is revised as “suggesting that the ability of different light treatment for maintaining organic acid content largely depends on the type of organic acid.”. see page 8, lines 218-220.

Point 39: Lines 239-240: "Since carotenoids play an important role in fruit before and after harvesting, it is vital to understand the..."The discussion on carotenoids is very weak.

Response 39: The sentence was deleted. The discussion was revised according to the suggestion. See pages 9-10, lines 228-294.

Materials and Methods

Point 40: Lines 253-255: As for "light (660 nm, 150 μmol m-2 s-1), blue light (470 nm, 200 μmol m-2 s-1), white light (100 μmol m-2 s-1), UVA (315-400 nm, 100 μmol m-2 s-1), UVB (270-315 nm, 100 μmol m-2 s-1), and UVC (100-280 nm, 100 μmol m-2 s-1)",

What was the rationale of these treatment levels?

Several important informations are missing in the description of treatment application:

What instrument was used for the measurement of light intensity?

Was the distance between the fruit and the light sources?

Were the fruit rotated during treatment?

Response 40: Thanks for pointing these out. According to the suggestion, the related information about light treatment was complemented in the revised manuscript. See “LEDs intensities were adjusted by the incubator, UV lamps have fixed light intensities, and all light intensities were confirmed by the spectrum analyzer (HR-350, Taiwan, China). The different types of light doses were obtained by the exposure at a fixed distance, and fruits were placed 20 cm below the lamps. During the treatment period, every eight hours, fruits were rotated vertically by 180° to ensure all fruits exposed to the same light irradiation.” in page 10, lines 308-316.

Point 41: Line 256: "…for six days." Were the fruit continuously subjected to these treatments over the entire 144 hours of the experiment? There was no period of darkness?

Response 41: See response to point 1.

Point 42: Line 261: I suggest revising "Chromatic aberration was" to "Color changes were".

Response 42: It was revised as "Color changes were ", see page11, lines 321.

Point 43: Line 263: "the sunny side and back side", This is not clear. Please revise and use a more common wording.

Response 43: It was revised as "the shaded and exposed side of each fruit were measured", see page11, lines 323-324.

Point 44: Line 267: "chromatic aberration", Replace by appropriate words.

Response 44: "chromatic aberration" was replaced by "color value", see page11, lines 327.

Point 45: Line 274: "Three grams of frozen segment cubes", Using directly the juice would have been more effective.

Response 45: Thanks for your good suggestion. In the study, frozen segment cubes were ground into a uniform powder, the powder was used to extract the soluble sugars and organic acids. The method is also stable and effective, and it is widely used in many fruit studies. The extraction procedure is different from using juice sample. I am not sure that the juice would have been more effective, maybe we will try the juice method in the future.

Point 46: Line 281: I suggest the sentence "Three milliliters of the supernatant was removed and then…"without the words " was blow", "a" and "blower".

Response 46: The sentence was revised as " Three milliliters of the supernatant was removed and then dried using nitrogen.", see page11, lines 333.

Point 47: Line 282: I suggest the sentence "Additionally, the dry residue on the wall…"without the words "on the wall".

Response 47: The words "on the wall" was removed from the sentence, see page11, line 340.

Point 48: Line 283: I suggest revising "purified using" to "passed through".

Response 48: "purified using" was revised "passed through" in the revised manuscript, see page 11, line 341.

Point 49: Line 287: "The detection was performed by as previously described", Give the information about the detectors used. Without this information it seems that PDA was also used for sugar.

Response 49: We sorry for our unclear description. In the study, a Waters 2998 Photodiode Array Detector was used to detect sugars and organic acids, the information was complemented in page 11, line 344.

Point 50: Line 292: I suggest the sentence "Three biological replicates were used for each sample point" without the word "point".

Response 50: The word "point" was removed from the sentence, see page 11, line 351.

Point 51: Lines 300: I suggest the sentence "Thereafter, the sample was blown dry with…"without the words "a" and "blower".

Response 51: The sentence was corrected as “Thereafter, the sample was dried with nitrogen”, see page 11, line 36.

Point 52: Lines 313-314: I suggest revising the sentence "The measured mobile phase of each sample was set as follows:" to "The elution program was set as follows:".

Response 52: The sentence was revised as “The elution program was set as follows:”, see page 12, line 374.

Point 53: Line 323: "triplicate experiments", This is not clear. Please revise.

Response 53: The sentence was revised as “Three biological replicates were used for all experiments”, see page 12, line 371.

Conclusions

Point 54: Line 329: I suggest the sentence "In this study, postharvest LEDs and UV light irradiation accelerated the fruit…"without the words "the fruit".

Response 54: "the fruit" was removed from the sentence, see page 12, line 386.

Point 55: Line 330: "in citrus fruit", This too broad. Please specify the name of the cultivar that was used.

Response 55: The specify the name of the cultivar was given “Hamlin” sweet oranges”, see page 12, line 387.

Point 56: Lines 331-333: I suggest the sentence "UV light can effectively induce accumulations of soluble sugars, blue light can effectively promote the increase of total organic acid and citric acid contents, and dark treatment and red light can effectively maintain the carotenoids contents." without the words "can effectively" and "can", and replacing "induce" by "induces" or "induced", "promote" by "promotes" or "promoted".

Response 56: Thanks for your suggestions. The induced" and "promoted" were used in the revised manuscript. See page 12, lines 388, 389.

Point 57: Lines 333-334: "Therefore, the effects of different types of light irradiation on fruit compounds were inconsistent…", This is not clear.

Response 57: The sentence was revised as “Red light and dark shade effectively promoted carotenoid accumulation in citrus fruit. Further research is needed to investigate the molecular mechanisms for the specific accumulation of different compounds.”, see page 12, lines 390-392.

Reviewer 3 Report

molecules-572129

Effect of light-emitting diode and ultraviolet irradiation on soluble sugars, organic acids and carotenoids of postharvest sweet orange (Citrus sinensis(L.) Osbeck)

Authors: Linping Hu, Can Yang, Lina Zhang, Jing Feng, Wanpeng Xi *

Submitted to section: Natural Products Chemistry,

 This manuscript describes evaluation the effect of postharvest LED and UV irradiation on basic ripening parameters. However, this manuscript is part of the experiment described in Molecules, 2019, 24, 1755 (not referenced) and is considered unsuitable for this journal. Therefore, it seems that there is a little lack of impact as a paper to be published in Molecules. I think you should refer to previous papers and submit to other journals.

Author Response

Comments and Suggestions for Authors:

This manuscript describes evaluation the effect of postharvest LED and UV irradiation on basic ripening parameters. However, this manuscript is part of the experiment described in Molecules, 2019, 24, 1755 (not referenced) and is considered unsuitable for this journal. Therefore, it seems that there is a little lack of impact as a paper to be published in Molecules. I think you should refer to previous papers and submit to other journals.

Response: Thanks for your comments. Yes, the study seems to be similar with our another work (Molecules, 2019, 24, 1755). However, even the light treatment is similar, in fact they are completely different independent experiment and study. Our previous study mainly focused on investigating the effect of light treatment on flavonoids and limonoids using ‘Newhall Navel Oranges’, and an efficient identification method (UPLC-qTOF-MS) was used in the study, which is one of the novelty of the study. Here, the present study mainly focused on investigating the effect of light treatment on fruit ripening, soluble sugars, organic acids and carotenoids using ‘Hamlin’ sweet orange. In addition, in our previous study, we found that flavonoids and limonoids responds differently to a single light source. Based on the point, we inferred whether the response of soluble sugars, organic acids and carotenoids to light treatment is similar with flavonoids and limonoids. To further verify our speculation, we designed the present study, and we draw the same conclusion, which provide important information with us to study the molecular mechanism of these crucial phytochemicals related citrus fruit quality. And these findings suggested that the process of light regulation role for flavonoids and carotenoids metabolism is very complex, even each compounds such as β-cryptoxanthin, zeaxanthin, and lutein is also different. These information is very important for future work.

  Compared with previous studies, the contribution and novelty of the present study is as follows:

LEDs and UV irradiation not only accelerated orange ripening but also caused significant changes in the contents of soluble sugars, organic acids and carotenoids. Soluble sugar, organic acid and carotenoid accumulation in sweet orange responded differently to UV and LEDs irradiation. Compared with harvest (day0), red light and dark shade treatment had a significant inducing effect on carotenoids accumulation, white light treatment had the most significant negative effect on carotenoid accumulation. Appropriate light irradiation is a potential effective way to maintain or improve the postharvest fruit quality.

Based on these points, these above mentioned findings is true different from our previous work (Molecules, 2019, 24, 1755), we think it is important for this field and our future work. And the novelty and importance of the study was put emphasis on the induction and discussion. See page 2, lines 62-64 and page 11, lines 299-303 without tracked changes.

Round 2

Reviewer 1 Report

The authors revised the manuscript according to my suggestion. However, for the result of carotenoid accumulation, I still cannot well understand. Why did the authors use the 0 day after harvest as the control? The main purpose of this study is to compare the effect of different lights. Thus, I think that the authors should choose dark treatment or white treatment as the control, and compared it with different lights on the same day after harvest.

Author Response

Comments and Suggestions for Authors:

The authors revised the manuscript according to my suggestion. However, for the result of carotenoid accumulation, I still cannot well understand. Why did the authors use the 0 day after harvest as the control? The main purpose of this study is to compare the effect of different lights. Thus, I think that the authors should choose dark treatment or white treatment as the control, and compared it with different lights on the same day after harvest.

Response: Thank you for pointing it out. After an in-depth consideration, we think you are right, because the contents of phytochemicals will change after harvest during storage without light treatment. So if we took the Day 0 as the control, the differences resulted from storage time and light treatment, but if we took the dark shade or white light, the real difference only resulted from light treatment. Based on the point, we accepted your suggestion, and now in the revised manuscript, we took the dark shade treatment as the control. All related descriptions were rewritten in Abstract (page 1, line 19) and Results section, all Figure legends and Materials and Methods (page 10, line 343), See the revised manuscript.

Reviewer 2 Report

The author has significantly improved the manuscript. However, further improvement is needed before acceptance can be recommended. Below are the main reasons for this recommendation.

1) The author fails to provide the rationale for selecting the light treatment levels used in this study.

2) It is odd that the author used a Photodiode Array Detector (PDA) for sugar analyses. More puzzling is the fact that the cited reference (Wit et al., 2016)  for this method is a review article on light-mediated hormonal regulation of plant growth and development. In this review article sugar analyses were not reported. This is an unacceptable case of wrong utilisation of the literature.

3) Another case of literature misuse is noted when the author used references 21 to 23 to support the statement " A good amount of progress has been made in UVB and UVC treatments;...". None of the cited references were studies on the impact of UV-B or UV-C on fresh fruits to maintain their postharvest quality. It is to be highlighted that the sentence quoted above needs edition of the English language.

3) For Figures 2, 3 and 4 the legend should give all the information that are needed for good understanding, as was done in Figure 1 regarding the statistical test that was conducted to separate the means.  Adding the number of replicates (n=...) will also be recommended.

4) There are still need for significant improvement of English language and style.

Author Response

Comments and Suggestions for Authors:

The author has significantly improved the manuscript. However, further improvement is needed before acceptance can be recommended. Below are the main reasons for this recommendation.

Point 1: The author fails to provide the rationale for selecting the light treatment levels used in this study.

Response 1: We are sorry for insufficient explanation. In our study, we investigated the effects of different types of light on the sugars, organic acids and carotenoids in the sweet fruits. Before designing the study, we investigated many previous studies, and confirmed the light treatment levels in our study.

For the red light, Ma et al. first reported in 2012 that irradiation with red light at intensity of 50 μmol m−2 s−1 for six days was effective in enhancing carotenoid contents, especially the content of β-cryptoxanthin, while blue LED light had no significant effect on the carotenoid content in the flavedo of Satsuma mandarin, which can be seen in Figure 1 of the reference (Ma et al. (2012). Journal of Agricultural and Food Chemistry, 60 (1), 197-201.). Then, in 2015, Ma et al. increased the intensity of the red LED light to 150 μmol m−2 s−1, and the results showed that the contents of β-cryptoxanthin, all-trans-violaxanthin, 9-cis-violaxanthin and lutein were simultaneously increased more along with the total carotenoid accumulation under red LED light, which can be seen in Figure 3 of the reference (Ma, G., et al. (2015). Postharvest Biology and Technology, 99(Supplement C): 99-104). Therefore, we chose red light of 150 μmol m−2 s−1 intensity in our present study.

For the blue light, based on Ma et al. first report in 2012, we inferred that the of intensity 50 μmol m−2 s−1 is not enough for citrus. As for a latter report, blue light for the intensity of 630 μmol m−2 s−1 for 18 hours increased the phenylpropanoids in citrus fruit (Ballester Frutos A R.et al. Journal of Agricultural and Food Chemistry 2017 218, 575-583). However, we found a previous report that the growth rate of microalgae Chlorella vulgaris can be accelerated most significantly by blue light with 200 μmol m−2 s−1 compared with both 100 and 300 μmol m−2 s−1, which can be seen in Figure 4, 6 and Table 1 of the reference (Atta, M., et al. (2013). Bioresource Technology, 148: 373-378.). Taking into account the experimental results and the difference in experimental processing time reported above, thus, we determined to set the blue light at 200 μmol m−2 s−1.

In addition, since there are few reports on the optimal intensity of other lights, we set the intensity of all ultraviolet and white lights as 100 μmol m−2 s−1, which is also the commonly used intensity of light irradiation treatment (Harbaum-Piayda et al. (2010). Postharvest Biology and Technology, 56(3): 202-208).

Based on these existing reports, we have combined the light levels that are commonly used in these experiments.

Point 2: It is odd that the author used a Photodiode Array Detector (PDA) for sugar analyses. More puzzling is the fact that the cited reference (Wit et al., 2016) for this method is a review article on light-mediated hormonal regulation of plant growth and development. In this review article sugar analyses were not reported. This is an unacceptable case of wrong utilisation of the literature.

Response 2: We are sorry for the error. When we readjusted the order of reference without endnote, the reference was cited incorrectly. In fact, soluble sugars were analyzed by high-performance liquid chromatography (HPLC), the chromatographic separation was conducted on a 0.5-μm NH2 column (4.6 mm × 250 mm, Shimadzu, Japan), and soluble sugars were detected with a 2424 evaporative light scattering detector (ELSD) (Waters Beckman Coulter Inc., Brea, CA, USA), see page 11, lines 375-379; The citation was revised, see page 8, line204. And the description was rewritten, see page 11, lines 375-381. In addition, we double-checked all citation in the manuscript.

Point 3: Another case of literature misuse is noted when the author used references 21 to 23 to support the statement " A good amount of progress has been made in UVB and UVC treatments;...". None of the cited references were studies on the impact of UV-B or UV-C on fresh fruits to maintain their postharvest quality. It is to be highlighted that the sentence quoted above needs edition of the English language.

Response 3: We are sorry for our wrong utilisation of the literature; the citation was also revised. Furthermore, the senescence was revised as "There have been a number of reports about the treatments of UVB and UVC exposure". See page 8, lines 207-208.

Point 4: For Figures 2, 3 and 4 the legend should give all the information that are needed for good understanding, as was done in Figure 1 regarding the statistical test that was conducted to separate the means.  Adding the number of replicates (n=...) will also be recommended.

Response 4: According to the suggestion, we have revised legends of Figures 1-4, see page 3, lines 89-92, page 4, lines 112-118, page 5, lines 134-140 and page 6, lines 161-167, respectively.

Point 5: There are still need for significant improvement of English language and style.

Response 1: Thank you for your pointing it out. We made our efforts to improve the English language, at the same time, we resubmit the manuscript to MDPI language service system to improve again by special revision. See the revised manuscript.